# A Cross-Sectional Analysis of Self-Medication Patterns during the COVID-19 Pandemic in Ecuador

**DOI:** 10.3390/medicina58111678

**Published:** 2022-11-19

**Authors:** Fabián Arias, Juan S. Izquierdo-Condoy, Patricio Naranjo-Lara, Verónica Alarcón, Paulina Bonilla, Elizabeth Erazo, Sarah J. Carrington, Esteban Ortiz-Prado

**Affiliations:** 1Faculty of Medical Sciences, Universidad Central del Ecuador, Quito 100201, Ecuador; 2One Health Research Group, Faculty of Health Science, Universidad de Las Americas, Quito 170507, Ecuador

**Keywords:** COVID-19, SARS-CoV-2, self-medication, drug use, Ecuador

## Abstract

*Background and Objectives*: Drug consumption is a widely developed practice around the world. However, sometimes medicines are acquired with or without prescription, a practice termed self-medication, which can have negative impacts on the health of the population. It has been observed that with the arrival of the coronavirus disease 2019 (COVID-19) pandemic, self-medicated drug consumption figures increased in several countries. To describe the patterns of medication, use and the prevalence of self-medication during the COVID-19 pandemic in inhabitants of the capital province of Pichincha, Ecuador. *Materials and Methods*: A descriptive, cross-sectional study was conducted based on a self-administered online questionnaire from April to June 2022, among residents of the province of Pichincha, Ecuador. Participants were invited through social networks (WhatsApp and Facebook). A total of 401 surveys were included in this study. Consumption patterns (prescription of and treatment with) of medicines during the pandemic were evaluated, as well as the prevalence of self-medication and variables that characterize the way of acquiring medicines. The Chi-square test was used to look for relationships between consumption patterns, self-medication, and the characteristics of the participants. *Results*: Most participants were female (53.4%), and 59.4% reported having had COVID-19. A total of 244 (60.9%) consumed medications during the pandemic, mostly for the purpose of treating the infection. About half (48.4%) self-medicated. The most used medications were paracetamol (87.3%) and ibuprofen (47.5%). Drugs consumption as a treatment and informal sources of information (TV, social networks, advice) were associated with the practice of self-medication (*p* < 0.05). *Conclusions*: A significant percentage of over-the-counter (OTC) and legal drug use was found to persist after the COVID-19 pandemic. Our findings highlight the effects that alternative forms of information sources other than medical personnel can have on drug consumption and self-medication practices.

## 1. Introduction

Coronavirus disease 2019 (COVID-19), produced by the severe acute respiratory syndrome coronavirus 2 (SARS-CoV-2 virus), is the cause of an infectious disease characterized by the presence of a wide range of symptoms and signs from mild to severe, sometimes even causing fatal outcomes [1].

COVID-19 first emerged in December 2019 in Wuhan, China. By January 2020, its rapid spread led the World Health Organization (WHO) to declare it a public health emergency of international importance, and by March 2020 it was declared a pandemic [1,2]. In Ecuador, in March 2020, the Government declared a national quarantine. According to official data from the Ministry of Public Health of Ecuador (MSP) until September 2022, approximately 1,005,521 confirmed cases and 35,899 deaths from COVID-19 have been reported [3].

The arrival of the COVID-19 pandemic in Ecuador caused an enormous feeling of uncertainty within the population, and, with respect to its adequate treatment. To this uncertainty was added substantial misinformation and a largely poor response by health services and governments globally. The combination of these factors translated into panic and confusion in the population. In this context, some Ecuadorians, feeling uncertainty as to the best treatment to reduce the harmful effects of the COVID-19 infection on their health, chose to carry out unconventional and scientifically unproven practices. While in some cases this included consuming mild medications without a medical prescription (self-medication), in others this extended to consuming various medications without proven efficacy and possibly even with harmful impacts. These included substances such as powerful antibacterial, antimalarials, anticoagulants, and substances without health registration such as chlorine hydroxide. These latter medications were often taken while ignoring the complications and adverse effects that its consumption could mean for health [4,5].

Self-medication is a growing problem worldwide and has shown to be influenced by various economic, sociodemographic, and cultural factors. An additional and significant driver of this problem emerged with the pandemic; however, as misinformation about COVID-19 treatments was disseminated not just through informal internet networks but even by conventional media outlets [6,7]. In Ecuador, as in most countries, analgesics, antipyretics, and antibiotics are the most frequently used medications without a medical prescription. Non-prescribed use of such medication brings significant risks to the population, such as drug interactions, overdosing, poisoning, and, in the case of antibiotics, bacterial resistance [8,9,10].

Research pertaining to the prevalence of self-medication during the COVID-19 pandemic, reveals that in low-income countries such as Peru, Bangladesh, Togo, and Nigeria, self-medication rates range from 34% to 84%, which is very high value. In countries such as Peru and Kenya, an increase in rates of 36.2% has been observed compared to the pre-pandemic period. Other countries for which data on the prevalence of self-medication is found are Bangladesh and Thailand, which report 88.3% and 88.2%, respectively [11].

In addition, complementary to the study of this problem, it has been proposed to demonstrate the differences between the practice of self-medication related to COVID-19 as prevention or management of the disease; however, the findings from a systematic review with robust methodology failed to explain in depth the characteristics of self-medication under this approach [12].

Although some studies have recognized the tendency to self-medicate in Ecuador [13], there have been no recent studies on self-medication in Ecuador, and none since the COVID-19 pandemic which is hypothesized to have heightened both the tendency and misuse of self-medication within the country. Given this, we designed this cross-sectional study performing an online survey.

The objective of the study is to better understand patterns of medication use and gage the prevalence of self-medication during the COVID-19 pandemic in inhabitants of the capital province of Pichincha, Ecuador.

## 2. Materials and Methods

### 2.1. Study Design

A descriptive, cross-sectional, province-wide study was performed using an online questionnaire.

### 2.2. Setting and Participants

Ecuador is a country located on the equator on the eastern most coast of the South American continent with a territory of approximately 283,560 km^2^ of surface. The Ecuadorian territory is divided into four geo-climatic regions: Coast, Andes (highlands), Amazonia and Galapagos Islands. Politically it is divided into 24 provinces. The province of Pichincha is located in Andes region, with a total population of 2,567,287 inhabitants according to the last population census, the majority (77.0%) of whom live in the Quito canton [14,15].

The study group consisted of residents from all the cantons of the province of Pichincha. Study participants were eligible to participate if they had internet access and were permanent residents in any of the cantons of the province of Pichincha. Consent was obtained from participants at the beginning of the questionnaire with an explanation of the purpose of the study. Participants could continue with the entire questionnaire only after giving consent (by electronically marking) a “Terms and Conditions” and “Participation Agreement” consent form.

### 2.3. Data Measurement and Questionnaire

A structured questionnaire was designed to evaluate the patterns of drug consumption and self-medication of the inhabitants of the province of Pichincha, Ecuador, during the COVID-19 pandemic. Initially, a pilot study was conducted with 30 participants close to the researchers with the opportunity to accept anonymous feedback to identify comprehension difficulties or errors in the structure of the questions that made up the original survey. The pilot study participants agreed to answer the questionnaire voluntarily and did not receive any positive or negative retribution for their participation to ensure the objectivity of the feedback observations.

After editing the errors detected in the pilot study, a 19-question questionnaire was created in Spanish, revised, and validated by an expert in Public Health. In addition, an English version of the questionnaire was designed and is presented in this manuscript. The construction of this version was carried out by one of the team’s researchers who is a native English speaker to ensure an accurate translation of the survey.

The final version of the online questionnaire consisted of three sections:

Section 1: one question for informed consent of participants participant.

Section 2: composed of eight questions regarding participant demographics, such as gender, age, marital status, occupation, and education.

Section 3: comprises ten questions, which the patterns (prevention and treatment) and characteristics of medication consumption of the participants since the beginning of the COVID-19 pandemic, as well as the practice of self-medication.

The Spanish-language questionnaire (the official language of Ecuador) was distributed through a single link belonging to the Google Forms platform via dissemination groups on social networks (WhatsApp and Facebook), from April 2022 to June 2022. In all cases, in the initial section of the questionnaire, a brief explanation was provided about the objective of the study, in addition to a section guaranteeing the confidentiality of the data collected. All questionnaires were anonymous, and no identifiable data were requested.

At the conclusion of the collection process, a total of 405 responses were recorded. For these responses, two members of the research group (JI and PN) oversaw reviewing the questionnaires to identify duplicate responses and questionnaires with clear contradictions. After the debugging process, 401 responses were accepted for analysis.

### 2.4. Study Size

According to official data, the total population between 16 to 65 years of age in Pichincha province was 1,452,262 inhabits [16]. From this population, the necessary sample size was calculated using the following equation [17]:$$n=\frac{\left(N\cdot Z^{2}\right)\cdot \;\left(p\cdot q\right)}{d^{2}\left(N-1\right)+Z^{2}\cdot \left(p\cdot q\right)}$$
where, the total population (N = 1,452,262), the expected proportion of unknown response was (*p* = 0.5), confidence level (Z = 95%, or Z = 1.96) and a precision or margin of error allowed of (d = 5%). A minimum of 385 participants was obtained.

### 2.5. Data Management

The evaluation of consumption patterns from the beginning of the pandemic was based on the characterization of the variable “cause of medicine consumption”. The variable allows for two response options: preventive consumption for cases in which medicines were consumed to prevent SARS-CoV-2 virus infection, and treatment consumption for cases in which medicines were consumed to treat symptoms or cure COVID-19 infection.

The “self-medication practice” variable evaluated participants’ methods of acquiring medicines based on the WHO definition of self-medication [18]. This variable allows for two response options: ethical consumption for participants who acquired medicines using a medical prescription or by receiving them from a health care facility, and consumption by self-medication for participants who acquired medicines without a medical prescription, through friends or relatives, and using medicines stored at home.

### 2.6. Statistical Methods

The descriptive analysis of the qualitative variables was carried out using frequencies and percentages. The Chi-square test was used to search for an association between qualitative variables. Values *p* < 0.05 were accepted as statistically significant. All the analysis of results was carried out in the IBM SPSS Statistics for Windows Version 24.0 software.

## 3. Results

### 3.1. General Results

A total of 401 responses were obtained. More than half were women (53.4%), the majority (79.1%) between 18 and 40 years old, 59.4% claimed to have had the COVID-19 infection, and 66.8% of respondents took medication during the pandemic. 51.6% of those who responded claimed to have taken these medications as a treatment for the infection, while 9.2% did so as prevention. Of those who consumed medications, 126 (51.6%) acquired the medications correctly (medical prescription and in hospital pharmacies), while 118 (48.4%) practiced self-medication (acquisition without prescription, through relatives or friends, or consuming medications that they had at home) (Table 1).

### 3.2. Medication Consumption Patterns

Most respondents who used medication during the pandemic reside in the city of Quito, where 84.3% claimed to have done so as a treatment for COVID-19 infection, while 15.7% as prevention (*p* = 0.062). Distinction by gender reveals that 83% of the men who used some medication did so for COVID-19 treatment whereas 86% of the women who did so, did for the same reason (*p* = 0.769) (Table 2).

Regarding the history of COVID-19, 95.7% of the participants who had COVID-19 stated that they had taken some medication as treatment, while 82.4% of the participants who did not have the infection stated that the reason for having used medication during the pandemic was for infection prevention (*p* < 0.001) (Table 2). In the same context, in general (as prevention or treatment), the medications most used by the participants were paracetamol (87.3%), and ibuprofen (47.5%). In the case of azithromycin, 91.4% (*n* = 85) used it as treatment for COVID-19 infection, and only 8.6% as prevention (*p* = 0.025) (Table 2).

The sources of information preferred by those who claimed to consume medications as treatment were medical prescription (94.0%), internet sites (66.7%) and social networks (56.5%), compared to those who claimed to consume them as prevention (*p* < 0.001) (Table 2). In another context, most participants who consumed medications as treatment did so through self-medication (72.9%) compared to those who did so as prevention (*p* < 0.001) (Table 2).

### 3.3. Self-Medication Analysis

Most participants who practiced self-medication were men (57.9%) (*p* = 0.015). Similarly, among the participants who had COVID-19 (*n* = 210), the majority (57.6%) reported having obtained the drugs under prescription from a medical professional (ethically), while those who mostly self-medicated had no history of COVID -19 (*p* < 0.001) (Table 3).

The group that reported having the highest number of adverse effects was the one that practiced self-medication (53.8%). However, the participants of this same group (self-medication) were the ones that most recommended the use (53.5%) of the drugs they consumed during the pandemic (*p* = 0.021) (Table 3).

Among the drugs studied, most were acquired in similar frequencies between the self-medication group and ethical acquisition; however, in the case of Azithromycin, most (72.0%) was acquired legally (*p* > 0.001) (Table 3).

The sources of information most used by those who practiced self-medication were informal sources (internet, (79.5%)), social networks (78.3%), TV/Radio (81.3%), and close friend/family advice (84.4%) (*p* < 0.05). Those who legally acquired the drugs stated that their main source of information was the prescription by a doctor (72.9%) (*p* < 0.001) (Table 3).

## 4. Discussion

The aim of this study was to describe the patterns of drug consumption and self-medication during the COVID-19 pandemic in the general population within the province of Pichincha, Ecuador.

We found that of the total number of participants, slightly more than half identified themselves as women (53.4%), a distribution similar to that observed in studies conducted by other authors in the context of self-medication and COVID-19 in different countries (Nigeria, Pakistan, India, Peru, China) [19,20,21,22,23]. We consider that this trend may be related to that reported by Smith W., who identified a significant positive association between the response rate to digital surveys and female gender [24]. Our findings indicated that, of the total number of participants, more than half (60.8%) consumed some type of medication during the outbreak. We felt it appropriate to exclude from the analysis those participants who reported regular medication use because this type of use was not triggered by the pandemic. Thus, we found that the largest proportion of remaining participants who reported taking medication during the pandemic did so as treatment (to cure or alleviate symptoms triggered by the infection) (51.6%), and a much smaller group as prevention (9.2%).

With respect to the analysis of consumption patterns, we identified important differences related to several descriptive characteristics of the participants. Initially, the highest proportions of those who reported having consumed drugs during the pandemic as treatment belonged to participants with higher levels of education, compared to those who reported having consumed drugs for preventive purposes. Within the same context, we observed that within the group of participants who had a history of COVID-19 infection, there was a higher proportion of drug use for treatment, while in those participants who reported not having this history there was a higher proportion of use for prevention (*p* < 0.001). Interestingly, fewer participants (*n* = 9) who consumed medication as prevention reported that they eventually became infected with COVID-19.

Another important aspect that we observed is that most of the medications (paracetamol, ibuprofen, azithromycin) were reported as being taken primarily for treatment of infection. On the other hand, most participants who reported having suffered adverse events (of any severity) due to medication use, belonged to the treatment group. Nevertheless, and curiously, a similar distribution was observed for those who would recommend the use of medications. In terms of information sources of drug users, we found that most participants who consumed medications as treatment, reported to have soured information about what to take through the internet, social networks, or prescription (*p* < 0.001). On the contrary, we observed that most participants who consumed medications as prevention, reported to have obtained information as to what to take through the radio or television (*p* < 0.001).

Although there was found to be significant differences in consumption patterns by consumption motivation, we are the first in the literature available to date to have identified consumption pattern differences by motivation for drug use during the pandemic. While this enables us to make an important contribution to self-medication drivers during a health crisis, we are unable to make an objective comparison of the findings presented with comparative studies. We can, however, compare the pandemic-specific self-medication trends to studies in several countries which have shown important differences by region (Nigeria, Poland, India, Togo, Peru, Pakistan) [19,20,21,22,25,26]. Quincho-Lopez et al. attempted to identify the characteristics of self-consumption in relation to COVID-19, although they obtained inconclusive results, most of their findings showed that the main characteristics for practicing self-medication around the treatment or management of COVID-19 infection are to alleviate symptoms, for limited access to health services and for giving importance to the opinion of those close to them [12]. Relatively, starting from the total number of responders (*n* = 401), in our study we found a relatively low overall prevalence of self-medication (29.4%). Yet, when specifically addressing the group of participants who reported consuming medications after the pandemic and not before (*n* = 244), we found a much higher percentage of self-medication (48.4%). Similar studies conducted in countries of this region have found similar self-medication figures. For example, in Peru, overall prevalence of self-medication was found to be 33.4%, and 54.8% amongst post pandemic only medication users [22,27].

At the time this research was carried out, vaccination rates against COVID-19 in Ecuador exceeded 80% coverage for the full schedule (two doses), which is why this characteristic was not studied in the participants. However, although the safety of vaccines has been widely demonstrated [28,29], the study of the possible influence of vaccination against COVID-19 on the self-medication practices of the population would be of interest to future research.

Disaggregation of the results by sex reveals that participants who identified themselves as men were shown to practice more self-medication practices during the pandemic compared to women (*p* = 0.015). Contradictory findings have been described in the literature regarding the influence of sex on self-medication behaviors. Nevertheless, in the research described by Sadio et al., in groups at high risk of infection (health personnel, air transport, police personnel, road transport and informal economic sectors) in Togo, it was reported that the practice of self-medication was significantly associated with being female [26]. A possible explanation given for these differences is that it could not rule out a possible influence of anxiety as a disturbing factor that is more prevalent in women, as has been described in populations in Iran and Italy [26].

In addition, we found a relationship with a higher proportion of self-medication in participants who reported taking medication for prevention compared to those who took it for treatment (*p* < 0.001). These differences have not been reported so far in other studies, although Okoye et al. find that the practice of self-medication was more common among participants who had a positive COVID-19 test (45.7%) compared to those who did not (29.7%) among health care professionals in three tertiary hospitals in southern Nigeria. This finding suggests a possible inclination of participants towards self-medication as treatment rather than prevention of COVID-19 infection [19].

The most frequently consumed medications by participants who self-medicated were paracetamol (87.2%), and ibuprofen (45.4%), followed by azithromycin (22%). Contradictorily, in a systematic review of the literature where the authors evaluated the prevalence and correlates of self-medication practices for the prevention and treatment of COVID-19 in 14 studies, they reported that (antibacterial) antibiotics (79%) were the most used agents in self-medication in all included studies, followed by vitamins (64%), antimalarials (50%), natural remedies (50%), and analgesics and antipyretics (43%) [30]. These differences between those and the present study could be explained by the existing restriction in Ecuador since the beginning of the pandemic related to the free sale of antibiotics, anti-influenza drugs and NSAIDs in pharmacies [31,32].

No significant differences were found between the frequency of consumption of the drugs studied (paracetamol, ibuprofen, hydroxychloroquine, ivermectin, penicillin, cold preparations) and the reason for use (prevention or treatment), except for azithromycin, which showed a related increase in its consumption as a treatment. This can be explained by the erroneous dissemination in formal and informal mass media, which developed from data from in vitro studies and with limited methodology in which it was assured that potent antimicrobials such as azithromycin, hydroxychloroquine and ivermectin had effects on the infection caused by the SARS-CoV-2 virus, and even had the capacity to prevent transmission [33,34,35]. In Ecuador, during the initial stages of the pandemic, the use of azithromycin was officially recommended as a treatment for COVID-19 in patients with severe cases without evidence of bacterial superinfection [36]. This recommendation was withdrawn months later from the official treatment regimens in light of new studies and the increased risk of serious adverse effects caused by the increase in consumption triggered by these recommendations [37]. Complementarily, in the case of azithromycin, the highest proportion of participants who claimed to have consumed it belonged to the group of “ethical acquisition” or correct acquisition of medicines. This indicates that the responsibility for consumption behaviors related to this antibacterial does not fall solely on the shoulders of the population and the media, but also partly upon those of Ecuadorian doctors who believed in the ability of this antibacterial against the SARS-CoV-2 virus and carried out prescribing practices of the same. Although in Ecuador there are no data related to antimicrobial prescribing practices in medical professionals during the COVID-19 pandemic, this situation has been documented by several authors in different countries. A study carried out in the United States investigated the prescription of antibiotics in older adults with outpatient management for COVID-19. Of the total participants, 29.6% received an antibiotic prescription, with azithromycin being the most frequently prescribed antibiotic (50.7%) [38]. In the same vein, in January of this year, the Italian Medicines Agency warned about a high shortage of azithromycin in that country, mainly due to inappropriate and excessive prescribing of this drug by health care providers, as patients can only obtain antibiotics from community or hospital pharmacies under prescription [39]. These circumstances remind us of the importance of optimizing antibiotic prescribing practices among health care personnel.

With respect to those participants who made use of self-medication options, the present study found that these individuals informed themselves of these medications more frequently through the internet, radio or television, social networks, and even accepting advice from people close to them (*p*-value < 0.05). These results contrast with those who did not resort to self-medication and were informed/received a medical prescription (*p* ≤ 0.001). In a study by Sanchez et al., of the general population in Spain, the results reveal that 27.4% of the study participants had consumed an over-the-counter (OTC) drug advertised on television and 58.5% of respondents reported using the internet to search for information about medications [40]. Similarly, in the Nigerian population, 54.3% of respondents claimed that television, radio, newspaper and social networks can influence self-medication behaviors with respect to COVID-19 [20].

In terms of patient-to-patient recommendation of medication, slightly more than half of the respondents who acquired medications through a prescription recommend this same behavior (57.6%), a higher proportion than the corresponding behavior within the self-medicated group (42.4%) (*p* = 0.021). While of a lower proportion, this latter figure remains significant. The prevalence of recommendation of self-medication practices may be the result of a lack of sufficient information about the risks of this apparently harmless behavior. It has been widely accepted that self-medication has a very important place in the health care system worldwide [18]. There are several factors that may interfere in this decision, such as saving time and money at the time of requesting a medical consultation, or reducing the loss of work time, among others. While these are perceived as advantages, it is also necessary to talk about the disadvantages that this practice may entail for people, the most representative being the health risks that arise from the appearance of adverse reactions, drug interactions, increased bacterial resistance and increased duration of the disease [41].

In Ecuador, a prescription issued by licensed health professionals is required for the sale of drugs to the public, except for over-the-counter drugs. Thus, they are marketed under two denominations, “OTC” and “prescription-only” [42]. Over the counter drugs maintain a risk/benefit margin that allows their use without medical supervision. In this way, the WHO has promoted responsible self-medication with these drugs as a form of self-care and which should not be categorized as an aberrant practice per se. Unfortunately, unlike in developed countries, in Ecuador “prescription-only” drugs are often sold without a prescription, as in other developing countries [43]. According to a report by Intercontinental Marketing Services Health (IMSH), as of 2010, 77% of patients in Ecuador purchase medicines without a prescription [42]. Different authors make a distinction between self-medication and self-prescription, the latter defined as the purchase or use of non-OTC or ethical drugs without a prescription prescribed by a professional [44]. From the beginning of the “outbreak” to date, it was mandated in the country that the sale of anti-flu, antibiotics and non-steroidal anti-inflammatory drugs be exclusively with a medical prescription [31]. Despite these measures, our findings showed that self-medication behavior had a significant presence during the COVID-19 pandemic. This may indicate the fact that these regulatory measures are insufficient, and that before pointing out the user as the main responsible for incorrect medication acquisition practices, the authorities, the pharmaceutical industry, and the media are also heavily implicated in this phenomenon.

Our study has several limitations inherent to the self-reported cross-sectional design. We cannot rule out the possibility that our findings are exposed to recall bias about the type of drugs consumed during the pandemic; however, we believe that the results collected may be very close to reality, because most of the drugs studied are over-the-counter and the general population is familiar with their names and use. On the other hand, since the questionnaire was distributed through social networks, information about the population that did not have the resources to access the questionnaire was left out of the study. Likewise, most older adults handle electronic devices and the internet with difficulty. This may cause selection bias; however, we believe that our sample calculation provided a considerable sample size to reduce this bias. Another limitation is that the results found in this research cannot be extrapolated to the entire Ecuadorian population since randomized sampling was not performed. Despite these limitations, our data were subjected to an extensive data filtering process to obtain valid results. For these reasons, we consider this study to be a reliable approximation of consumption and self-medication practices during the COVID-19 pandemic and may help in some way to propose improvements in the process of acquisition and consumption of medications in Ecuador.

## 5. Conclusions

A significant percentage of drug consumption was observed from the beginning of the COVID-19 pandemic, with an important proportion of self-medication practices observed within the residents of the capital province of Pichincha. The most consumed drugs were OTC drugs (paracetamol and ibuprofen); however, there was also a significant consumption of azithromycin. The most common cause of drug consumption was as a treatment for COVID-19 infection, which was associated with higher consumption of azithromycin. On the other hand, the practice of self-medication was associated with male sex and untrained sources of information. The consumption and self-medication figures showed a considerable influence of the pandemic on the practices of the participants. Our results highlight the effects of information sources as well as medical personnel on medication consumption and self-medication practices. We hope that the present study’s results will inform the health authorities and actors about action spaces to improve the ethical and OTC drug consumption behaviors in the Ecuadorian population.

## Figures and Tables

**Table 1 medicina-58-01678-t001:** Demographic characteristics of the participants.

Characteristics		*n*	(%)
Sex	Male	186	46.4%
	Female	214	53.4%
	Prefer not response	1	0.2%
	Total	401	100%
Residence	Quito	368	91.8%
	Rumiñahui	13	3.2%
	Mejía	7	1.8%
	Cayambe	10	2.5%
	Pedro Moncayo	2	0.5%
	Pedro Vicente Maldonado	1	0.2%
	Total	401	100%
Age (years)	Less than 18	5	1.2%
	18 to 40	317	79.1%
	41 to 65	77	19.2%
	More than 65	2	0.5%
	Total	401	100%
Civil Status	Single	245	61.1%
	Married	103	25.7%
	Divorced	29	7.2%
	Civil union	23	5.8%
	Widowed	1	0.2%
	Total	401	100%
Scholarship	Basic	2	0.5%
	High School Degree	96	23.9%
	College Degree	261	65.1%
	Master’s Degree	42	10.5%
	Total	401	100%
Occupation	Health care workers	78	19.4%
	Students	70	17.5%
	Indoor Activities	190	47.4%
	Outdoor Activities	63	15.7%
	Total	401	100%
COVID-19 History	Yes	238	59.4%
	No	163	40.6%
	Total	401	100%
COVID-19 Diagnostic	Laboratory tests	169	71.0%
	Clinical diagnosis	9	3.8%
	Close transmission	34	14.3%
	Symptoms without test	26	10.9%
	Total	238	100%
Medication use	Yes	268	66.8%
	No	133	33.2%
	Total	401	100%
Reason for use	Regular use	24	6.0%
	Prevention	37	9.2%
	Treatment COVID-19 infection	207	51.6%
	Not use	133	33.2%
	Total	401	100%
Acquisition means	Prescription	114	46.7
	Without prescription	74	30.3
	Parent/Friend	9	3.7
	Hospital	12	4.9
	Home	35	14.3
	Total	244	100%

**Table 2 medicina-58-01678-t002:** Consumption pattern distribution according to the participant’s characteristics.

Characteristics		Prevention		Treatment			
		*n*	%	*n*	%	Total	*p* Value
Residence	Quito	34	15.7	182	84.3	216	0.062
	Rumiñahui	0	0.0	11	100.0	11	
	Mejía	0	0.0	5	100.0	5	
	Cayambe	1	11.1	8	88.9	9	
	Pedro Moncayo	1	50.0	1	50.0	2	
	Vicente Maldonado	1	100.0	0	0.0	1	
Sex	Male	19	16.7	95	83.3	114	0.769
	Female	18	14.0	111	86.0	129	
	N/A	0	0.0	1	100.0	1	
Age (years)	Less than 18	0	0.0	4	100.0	4	0.081
	18 to 40	27	14.3	162	85.7	189	
	41 to 65	9	18.0	41	82.0	50	
	More than 65	1	100.0	0	0.0	1	
Civil Status	Single	18	12.6	125	87.4	143	0.085
	Married	10	15.9	53	84.1	63	
	Divorced	3	13.6	19	86.4	22	
	Civil Union	6	40.0	9	60.0	15	
	Widowed	0	0.0	1	100.0	1	
Scholarship	Basic	0	0.0	1	100.0	1	0.514
	High School Degree	12	20.7	46	79.3	58	
	College Degree	22	14.2	133	85.8	155	
	Master´s Degree	3	10.0	27	90.0	30	
COVID-19 History	Yes	9	4.3	201	95.7	210	**<0.001**
	No	28	82.4	6	17.6	34	
Acquisition means	Self-medication	32	27.1	86	72.9	118	**<0.001**
	Legally acquired	5	4.0	121	96.0	126	
Recommendations about medication	Yes	32	15.8	170	84.2	202	0.517
	No	5	11.9	37	88.1	42	
Adverse effects	Yes	6	23.1	20	76.9	26	0.234
	No	31	14.2	187	85.8	218	
Drugs used							
Paracetamol	Yes	30	14.1	183	85.9	213	0.218
	No	7	22.6	24	77.4	31	
Ibuprofen	Yes	16	13.8	100	86.2	116	0.57
	No	21	16.4	107	83.6	128	
Azithromycin	Yes	8	8.6	85	91.4	93	**0.025**
	No	29	19.2	122	80.8	151	
Hydroxychloroquine	Yes	0	0.0	3	100.0	3	0.461
	No	37	15.4	204	84.6	241	
Ivermectin	Yes	5	17.9	23	82.1	28	0.673
	No	32	14.8	184	85.2	216	
Penicillin	Yes	0	0.0	8	100.0	8	0.224
	No	37	15.7	199	84.3	236	
Flu and cold preparations	Yes	1	9.1	10	90.9	11	0.566
	No	36	15.5	197	84.5	233	
Information sources							
Internet	Yes	13	33.3	26	66.7	39	**0.001**
	No	24	11.7	181	88.3	205	
TV/Radio	Yes	9	56.3	7	43.8	16	**<0.001**
	No	28	12.3	200	87.7	228	
Social networks	Yes	10	43.5	13	56.5	23	**<0.001**
	No	27	12.2	194	87.8	221	
Advice from someone close	Yes	11	17.2	53	82.8	64	0.599
	No	26	14.4	154	85.6	180	
Prescription	Yes	10	6.0	156	94.0	166	**<0.001**
	No	27	34.6	51	65.4	78	

*p* values were calculated from Chi-square test.

**Table 3 medicina-58-01678-t003:** Self-medication distributions according to the participant’s characteristics.

Characteristics		Self-Medicated		Ethically Acquired			
		*n*	%	*n*	%	Total	*p* Value
Residence	Quito	107	49.5	109	50.5	216	**0.044**
	Rumiñahui	1	9.1	10	90.9	11	
	Mejía	2	40.0	3	60.0	5	
	Cayambe	6	66.7	3	33.3	9	
	Pedro Moncayo	2	100.0	0	0.0	2	
	Vicente Maldonado	0	0.0	1	100.0	1	
Sex	Male	66	57.9	48	42.1	114	**0.015**
	Female	52	40.3	77	59.7	129	
	N/A	0	0.0	1	100.0	1	
Age (years)	Less than 18	3	75.0	1	25.0	4	0.459
	18 to 40	92	48.7	97	51.3	189	
	41 to 65	22	44.0	28	56.0	50	
	More than 65	1	100.0	0	0.0	1	
Civil Status	Single	74	51.7	69	48.3	143	0.144
	Married	28	44.4	35	55.6	63	
	Divorced	6	27.3	16	72.7	22	
	Civil union	9	60.0	6	40.0	15	
	Widowed	1	100.0	0	0.0	1	
Scholarship	Basic	0	0.0	1	100.0	1	0.788
	High School Degree	29	50.0	29	50.0	58	
	College Degree	74	47.7	81	52.3	155	
	Master´s Degree	15	50.0	15	50.0	30	
COVID-19 History	Yes	89	42.4	121	57.6	210	**<0.001**
	No	29	85.3	5	14.7	34	
Adverse effects	Yes	12	46.2	14	53.8	26	0.812
	No	106	48.6	112	51.4	218	
Recommendations about medication	Yes	94	46.5	108	53.5	202	**0.021**
	No	24	57.1	18	42.9	42	
Drugs used							
Paracetamol	Yes	103	48.4	110	51.6	213	0.997
	No	15	48.4	16	51.6	31	
Ibuprofen	Yes	56	48.3	60	51.7	116	0.981
	No	62	48.4	66	51.6	128	
Azithromycin	Yes	26	28.0	67	72.0	93	**<0.001**
	No	92	60.9	59	39.1	151	
Hydroxychloroquine	Yes	1	33.3	2	66.7	3	0.601
	No	117	48.5	124	51.5	241	
Ivermectin	Yes	11	39.3	17	60.7	28	0.307
	No	107	49.5	109	50.5	216	
Penicillin	Yes	4	50.0	4	50.0	8	0.925
	No	114	48.3	122	51.7	236	
Flu and cold preparations	Yes	6	54.5	5	45.5	11	0.674
	No	112	48.1	121	51.9	233	
Information sources							
Internet	Yes	31	79.5	8	20.5	39	**<0.001**
	No	87	42.4	118	57.6	205	
TV/Radio	Yes	13	81.3	3	18.8	16	**0.006**
	No	105	46.1	123	53.9	228	
Social networks	Yes	18	78.3	5	21.7	23	**0.003**
	No	100	45.2	121	54.8	221	
Advice from someone close	Yes	54	84.4	10	15.6	64	**<0.001**
	No	64	35.6	116	64.4	180	
Prescription	Yes	45	27.1	121	72.9	166	**<0.001**
	No	73	93.6	5	6.4	78	

*p* values were calculated from Chi-square test.

## Data Availability

Not applicable.

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
