# Peer review of "A Cross-Sectional Analysis of Self-Medication Patterns during the COVID-19 Pandemic in Ecuador"

_medicina, 2022, doi:10.3390/medicina58111678_

Round 1

Reviewer 1 Report

The manuscript under review is a cross-sectional study that evaluates the type of self-medication used by people during the SARS-CoV-2 pandemic. The topic is interesting, and the research well conducted. I only have a few comments:

-          In the abstract, I think the use of “legal drug” instead of “drug” is meaningless because the it is clear that the paper is about pharmacological therapy and not abuse drug. I would modify the first abstract sentences (lines 9-11).

-          In the materials and methods, the authors should state the time period in which the questionnaire have been proposed to the people (for example from June 2020 to June 2022).

-          Line 190, what do the authors mean with “obtained the medications ethically”? legally? If so, please be clearer.

-          Another limitation of the study is the possibility of recall bias because people may not remember, or not remember properly, the kind of medications consumed. In the discussion, the authors should add this limit (lines 363-374).

-          I think the reference list should be implemented with the following: 10.3390/vaccines10020308; 10.1371/journal.pone.0259317

-          It would be interesting to evaluate if people that took preventive drugs for COVID then got vaccinated. Do the Authors also have such data? If not, it may be in the future another question for further research.

Reviewer 2 Report

Very interesting topic, the researchers did a great job to better understand patterns of medication use and gage the prevalence of self-medication during the COVID-19 pandemic. However, the article could be strengthened through:

Line 34: Introduce every acronym before using it in the text. The first time you use the term, put the acronym in parentheses after the full term, like coronavirus disease of 2019 and SARS-CoV-2 virus. As well as OTC.

Line 41-42: Update the confirmed cases before publication.

Add more towards scope of the problem in introduction section.

Add objectives of the study at the end of introduction.

What measures did you use to insure accurate translation of the tool?

Validity and reliability for the study and the pilot study need to be addressed.

Good luck
